# Effect of Inorganic Phosphorus Manipulation on the Growth and Progression of Prostate Cancer Cells In Vitro

**DOI:** 10.3390/ijms26104762

**Published:** 2025-05-16

**Authors:** Christina Mary Kourie, Fatima Ghamlouche, Sana Hachem, Amani Yehya, Layal Jaafar, Carla El-Mallah, Wassim Abou-Kheir, Omar Obeid

**Affiliations:** 1Department of Nutrition and Food Science, Faculty of Agriculture and Food Sciences, American University of Beirut, Beirut P.O. Box 11-0236, Lebanon; ck74@aub.edu.lb (C.M.K.); lsj08@mail.aub.edu (L.J.); cae14@mail.aub.edu (C.E.-M.); 2Department of Anatomy, Cell Biology and Physiological Sciences, Faculty of Medicine, American University of Beirut, Beirut P.O. Box 11-0236, Lebanon; fig00@mail.aub.edu (F.G.); sih12@mail.aub.edu (S.H.); ay46@aub.edu.lb (A.Y.)

**Keywords:** inorganic phosphorus, prostate cancer, in vitro, epithelial–mesenchymal transition, PLum-AD murine PC cells

## Abstract

Epidemiological studies have linked higher serum and dietary phosphorus to an increased risk of prostate cancer (PC) and its lethal state. However, these findings do not distinguish between the impact of inorganic phosphorus (Pi) *per se* and the impacts of its homoeostatic regulators. Thus, this study aimed to determine the in vitro tumorigenic effects of elevated Pi concentrations *per se* on androgen-dependent epithelial-like PLum-AD murine PC cells at molecular and cellular levels. Physiologically attainable elevated levels and supraphysiological levels of sodium (NaPi) and potassium phosphate (KPi) were used to assess PLum-AD cell proliferation, viability, migration, and epithelial–mesenchymal transition (EMT) marker expression, which were determined by the thiazolyl blue tetrazolium bromide cell assay, trypan blue exclusion assay, wound healing assay, and immunofluorescence staining, respectively. Treatment of Plum-AD cells with supraphysiological levels of NaPi (20 mM) significantly reduced cell proliferation, whereas KPi did not, suggesting a potential sodium-dependent Pi uptake mechanism. Furthermore, physiologically relevant elevated concentrations of NaPi (3 mM) and KPi (1 and 3 mM) increased the relative vimentin expression of PLum-AD PC cells, a biomarker of EMT. Our findings suggest that elevated Pi levels *per se*, in the hyperphosphatemia range, can directly promote EMT in PC, highlighting the potential role of Pi in tumor progression.

## 1. Introduction

Prostate cancer (PC) was ranked as the second most frequently diagnosed cancer and the fifth leading cause of cancer-related deaths among males worldwide in 2020, with an estimated 1.4 million new cases and 375 thousand deaths, respectively [1]. Patients with localized PC have a nearly 100% five-year survival rate due to a slow tumor growth rate, while those diagnosed with high-risk PC experience a poor survival rate of less than 30% due to rapid progression and metastasis, highlighting the need to address the current rising burden of advanced PC [2,3,4,5,6,7]. Although the exact molecular mechanisms driving PC carcinogenesis and progression remain unclear, the transformation of the prostate gland’s benign epithelial cells into malignant phenotypes is thought to stem from genetic and epigenetic alterations [8]. Progression from the first stage of tumor development to the metastatic stage is characterized by a decrease in cells expressing cytokeratin 8 (CK8), cytokeratin 18 (CK18), and the tumor suppressor gene p63. This progression is driven by the activation of genes regulating the epithelial–mesenchymal transition (EMT) pathway, consequentially involving the repression of epithelial markers like E-cadherin and the upregulation of mesenchymal markers such as vimentin (Vim) [9,10]. The risk factors for PC include both non-modifiable and modifiable factors. Non-modifiable risk factors include advanced age, specific genetic mutations, ethnicity, and family history [11]. Modifiable risk factors include diet and lifestyle [12]. Various studies have emphasized the potential role of minerals, such as selenium and zinc, in modulating PC prognosis [13,14], highlighting the importance of minerals in influencing PC tumorigenesis [15]. Interestingly, growing evidence have associated increased serum levels and intakes of phosphorus with an increased risk of PC and its advanced lethal stage [16].

Phosphorus is an essential mineral for various biological functions like nucleic acid production, cell signaling, and energy transfer. The serum concentration of inorganic phosphorus (Pi) is tightly regulated at between 0.8 and 1.45 mmol/L [17,18], and hyperphosphatemia is defined as a serum Pi exceeding 1.45 mmol/L, while severe hyperphosphatemia is defined as levels greater than 4.52 mmol/L [19]. The homeostatic regulation of phosphorus is highly controlled at the level of the kidney [20], meaning the fasting serum level is a weak indicator of intake [21,22]. Nonetheless, diurnal variations in serum Pi exist and these are related to the postprandial changes following meal ingestion. The magnitude of these changes is up to 0.5 mmol/L and mainly dependent on the meal content of phosphorus and carbohydrate [23,24,25]. Interestingly, excreting 1500 mg/day of dietary phosphorus was reported to take up to 20 h, suggesting a temporary storage pool in the intracellular fluid (ICF), as higher doses were shown to normalize serum levels within 8 h [26].

Beyond its role in fundamental physiological processes, phosphorus has been increasingly recognized as a signaling molecule capable of modulating tumor cell behavior. Several types of cancer cells have been reported to overexpress transmembrane phosphate transporters that are known to enhance phosphate uptake from the tumor microenvironment (TME) [27,28,29]. Indeed, inhibiting a sodium-dependent phosphate transporter in human lung cancer tissues led to suppressed tumor growth, decreased cell proliferation, and increased apoptosis [30]. Nonetheless, Bobko et al. [31] reported elevated Pi levels in the TME of metastatic, compared to non-metastatic breast tumors in a mouse model, identifying Pi as a TME marker for tumor progression. Moreover, serum Pi levels were more than 2-fold higher in patients with head, lung, neck, or cervical cancers compared to healthy individuals [32]. Excess intracellular Pi has been proposed to increase the metabolic activity of tumors, induce growth-promoting cell signaling, and potentially promote metastasis [33]. The upregulation of phosphate transporters and sequestration of excess Pi by cancer cells aligns with the Growth Rate Hypothesis, which suggests that tumor cells have a high demand for Pi to support their accelerated growth and energy needs, particularly for ribosome and RNA synthesis [34]. Several preclinical studies have demonstrated that elevated Pi levels can promote cancer cell proliferation, survival, tumor growth, and metastasis across various models, including lung, skin, and breast cancers [35,36,37]. These tumor-promoting effects are reported to be, at least in part, mediated by the activation of several key signaling oncogenic pathways such as the phosphatidylinositol 3-kinase/protein kinase B (PI3K/Akt) pathway and the mitogen-activated protein kinase/extracellular signal-regulated kinase (MAPK/ERK) pathway [35,36,38]. Despite these advancements, the specific molecular targets and subsequent downstream pathways resulting from elevated Pi exposure in PC remain unexplored.

Epidemiological data have suggested that a high dietary phosphorus intake is associated with an increased risk for PC [16,39,40]. Specifically, elevated serum phosphate and an increased dietary phosphorus intake were linked to a 7% and 8% increase in PC risk, respectively [13]. Additionally, results from a Mendelian randomization (MR) study indicated that genetically predicted high serum phosphorus was associated with a 19% higher PC risk per standard deviation increase [16]. Data from the 2020 UK Biobank support these findings, showing an association between circulating phosphate levels and an increased PC risk [39,40]. In addition to its well-established role in energy metabolism, phosphorus is closely linked to the regulation of the active form of vitamin D—calcitriol, parathyroid hormone (PTH), and fibroblast growth factor 23 (FGF23), which are factors that have been implicated in PC tumorigenesis [41,42,43]. A high phosphorus intake has been shown to increase circulating levels of FGF23 [44]. Elevated FGF23 subsequently suppresses the renal synthesis of calcitriol, which has been shown to help mitigate the risk and progression of PC [16,44]. Moreover, FGF23 expression and activity have been associated with PC initiation and progression [42,45]. Furthermore, increased serum Pi concentrations also stimulate the release of PTH from the parathyroid gland [44]. PTH has been identified as a mitogen for PC cells, with studies reporting that it enhances the proliferation and migration of PC cells in vitro and in vivo [46]. Nonetheless, the Health Professionals Follow-Up Study demonstrated that a higher dietary phosphorus intake is independently associated with an increased risk of lethal and high-grade prostate cancer, even after adjusting for other risk factors such as calcium intake [40]. Although these studies support an association between phosphorus (both serum levels and dietary intake) and PC, the underlying mechanisms remain unclear. Therefore, the current work focuses on determining the effects of elevated phosphorus on the growth and progression of PC cells independent of hormonal factors such as PTH, FGF23, and vitamin D.

Thus, we studied the short-term tumorigenic effects of elevated extracellular Pi concentrations *per se* using a 2D murine early-stage prostate cancer model at both the cellular and molecular levels. To this end, PC cells were treated with two different types of Pi for up to 72 h at varying concentrations to reflect normal serum levels, physiologically relevant hyperphosphatemia, and severe hyperphosphatemia.

## 2. Results

### 2.1. Effect of Pi on Cell Proliferation of PLum-AD Cells

Cells treated with increasing concentrations of sodium phosphate (NaPi) and potassium phosphate (KPi), ranging from 1 to 20 mM for up to 72 h, were assessed for proliferation using the 3-(4, 5-dimethylthiazol-2-yl)-2, 5-diphenyltetrazolium bromide (MTT) assay. Our data showed that cell proliferation was not significantly enhanced upon treatment with 1, 3, 6, or 9 mM NaPi or KPi (Figure 1a,b). However, at 24 h post-treatment, a significant reduction in cellular proliferation was observed in cells treated with 20 mM NaPi (*p* < 0.05) but not 20 mM KPi (Figure 1a,b). A similar trend upon treatment with 20 mM NaPi and KPi was observed at 48 and 72 h, though it was not statistically significant. Given that such levels are unlikely to occur physiologically, even in rare instances, we did not explore their effects in further experimental procedures. Importantly, no significant changes in cellular proliferation were detected in cells treated with sodium chloride (NaCl) or potassium chloride (KCl) (Figure 1b,c), implying that the cell proliferation inhibitory effect is unrelated to sodium or potassium.

### 2.2. Effect of Pi on Cell Viability of PLum-AD Cells

Cells treated with increasing concentrations of NaPi and KPi (1–9 mM) for 24, 48, and 72 h were assessed for viability using the trypan blue assay. No significant changes in cell viability of PLum-AD cells were observed at any concentration or time point upon treatment with NaPi, KPi, or KCl (Figure 2a,b,d). However, treatment with 3 and 6 mM NaCl significantly increased the viability at 24 h (*p* < 0.01), while treatment with 9 mM NaCl significantly decreased the cell viability at 72 h (*p* < 0.01) (Figure 2c). Since supraphysiological hyperphosphatemia (≥6 mmol/L) is very rare, further experiments focused on treating cells with lower concentrations.

### 2.3. Effect of Pi on the Migratory Ability of PLum-AD Cells

The migratory ability of PLum-AD cells in response to 1 and 3 mM NaPi and KPi treatment was assessed using the wound healing assay. Analysis of the percentage of open wound areas showed no significant differences between the treatment groups, with all the conditions achieving complete wound closure after 14 h (Figure 3a). Our results demonstrate that treatment with 1 mM NaPi or KPi, reflecting physiologically relevant hyperphosphatemia, and 3 mM NaPi or KPi, reflecting severe hyperphosphatemia, does not affect the migratory capacity of PLum-AD cells (Figure 3b,c).

### 2.4. Effect of Pi on Vimentin Expression in PLum-AD Cells

Treatment of PLum-AD cells with 1 or 3 mM NaPi and KPi for 24 and 48 h did not significantly affect CK8 expression. However, treatment with 3 mM NaPi and 1 and 3 mM KPi significantly increased the relative expression of Vim at 24 h post-treatment (*p* < 0.05 and *p* < 0.01, respectively). A similar trend was observed at 48 h following treatment with both concentrations of NaPi and KPi; however, the increase in relative vimentin expression did not reach statistical significance (Figure 4c,d). These findings suggest that physiologically relevant and severely elevated concentrations of NaPi and KPi enhance the expression of Vim, a biomarker for EMT, in PLum-AD cells.

## 3. Discussion

The current in vitro study investigated the effect of Pi *per se* on the growth and progression of PC, with a focus on utilizing Pi concentrations within a range attainable in humans.

The observed reduction in cell proliferation of PLum-AD cells treated with 20 mM NaPi aligns with other researchers’ findings. Incubation of human embryonic kidney 292 (HEK 293) and cervical cancer (HeLa) cells with 16 mM of NaPi and above was reported to trigger apoptosis [47], while others observed apoptosis starting at an even lower concentration (>2.5 mM NaPi) in endothelial cells [48]. Similarly, 20 mM of Pi in various forms (NaH_2_PO_4_, Na_2_HPO_4_, and KH_2_PO_4_) induced toxic effects on triple-negative human breast cancer cells (MDA-MB-231 cells) [49]. Interestingly, however, Spina et al. [50] demonstrated that 2.5–10 mM Pi inhibited proliferation in MDA-MB-231 cells, but had no effect on non-triple-negative breast cancer cells. These controversial findings suggest that Pi may have a differential effect on cancer cell tumorigenesis depending on the type and molecular profile of the cancer cells.

On the other hand, the failure of comparable concentrations of KPi, NaCl, and KCl to reduce cell proliferation indicates that the observed differential effect upon treatment with 20 mM NaPi may not be related to potential changes in osmolarity or the individual impact of Na or Pi but may be related to the possible presence and role of Na-dependent P transporters. Na-dependent Pi transporters, such as sodium-dependent phosphate co-transporter 2b (NaPi2b), are reported to play an important role in cancer metastasis. Inhibiting Na-dependent Pi transporters induced the mesenchymal–epithelial transition (MET) in MDA-MB-231 cells, thus reducing their migration ability [51]. Similarly, inhibiting NaPi2b in a K-ras mutated mouse model of lung cancer resulted in suppressed tumor growth, reduced cell proliferation, and enhanced apoptosis [30]. These transporters are overexpressed in 90% of epithelial ovarian cancers and other malignancies such as breast, thyroid, and lung cancers [27,28,29]. NaPi2b has also emerged as a potential predictive marker for targeted ovarian cancer therapy [52]. Thus, identifying and characterizing Pi transport mechanisms in PC cell lines could help elucidate their role in PC tumorigenesis and potentially serve as a way to target the uptake of Pi by PC cells. Nonetheless, the reduction in cell proliferation may not be relevant in vivo as it appeared at a concentration far above attainability in humans.

Cell viability and cell proliferation of non-cancerous and cancerous cell lines were reported to increase with an increased concentration of NaPi, and this was seen even at physiological concentrations. He et al. [47] found that increased extracellular NaPi (up to 10 mM) enhanced cell viability and the proliferation of human HEK 293 and HeLa cells. Likewise, treatment with increasing Pi concentrations (1–10 mM) accelerated the growth of human lung cells through the activation of the Akt and MAPK signaling pathways [36]. Camalier et al. [35] demonstrated that physiological increases in phosphate could promote cell proliferation in an epidermal cell line through the activation of ERK1/2 and Akt. However, our data showed that the proliferation and viability of PLum-AD cells were not enhanced by physiologically relevant or severe concentrations of Pi in the form of NaPi and KPi. Further investigations are needed to elucidate the mechanisms underlying these variations, particularly the involvement of the Akt, MAPK, and ERK1/2 signaling pathways.

Similarly, our results showed that Pi (NaPi or KPi) concentrations of 3 mM and below did not affect the migratory ability of PLum-AD cells, contrary to findings by others [37,53]. Cell migration and adhesion were stimulated in MDA-MB-231 cells upon treatment with elevated concentrations of Pi [53]. The discrepancy observed between our study and the study by Lin et al. [37] may stem from the differences in the models used. While Lin et al. [37] studied the effect of elevated Pi on the migratory abilities of a non-cancerous endothelial cell model in the presence of cancerous cells, our study determined the effect of elevated Pi concentrations on a malignant cell line already with a high basal migration rate. Additionally, PLum-AD cells migrated toward the scratch area, completely closing the wound by 14 h post-Pi treatment, potentially before Pi could mediate its intracellular effects.

On the other hand, the relative expression of the EMT marker Vim was upregulated following treatment with physiologically relevant elevated concentrations of NaPi and KPi. Specifically, 1 mM KPi and 3 mM NaPi induced a 1.7-fold increase in Vim expression, while 3 mM KPi resulted in a 2-fold increase compared to the control, suggesting that elevated Pi concentrations, relative to normal serum levels, can promote EMT in PC. Similarly, recent findings showed that supraphysiological Pi levels enhance EMT by suppressing E-cadherin while upregulating Vim expression in HEK 293 and HeLa cells [33]. Additionally, chemical or genetic inhibition of Pi transporters was shown to prevent high-Pi-mediated EMT of HEK 293 and HeLa cells [47].

While physiologically elevated concentrations of Pi did not affect proliferation, viability, or migratory abilities of PLum-AD PC cells, our findings suggest that elevated Pi in the TME *per se* increased Vim expression, thereby promoting EMT. These effects are independent of the regulatory endocrine factors such as FGF-23, calcitriol, and PTH, which were suggested to affect PC tumorigenesis [42,43,54].

A key strength of this study is the use of the PLum-AD cell line, which harbors *Pten* and *TP53* deletions, enabling self-renewal, growth into castration-resistant prostate cancer (CRPC) cells, and EMT-driven metastasis [55]. This model provides valuable insights into the tumorigenic effects of elevated Pi concentrations. Additionally, we used a spectrum of Pi concentrations, representing the physiological range seen in human hyperphosphatemia, which enhances our findings’ clinical relevance. However, a limitation of this study is the lack of investigation into the modulation of key signaling pathways known to promote EMT, such as Akt, MAPK, and ERK1/2, in response to elevated Pi levels in PLum-AD PC cells. Future studies should explore these downstream signaling pathways and use targeted inhibitors against them to strengthen the mechanistic link between elevated Pi-induced EMT and PC metastasis. Additionally, the in vitro model does not replicate the complexity of the in vivo TME. Thus, further investigations employing in vivo models are warranted to validate the present in vitro findings and assess the long-term impact of elevated Pi on PC progression and metastasis. Nonetheless, studying the endocrine-mediated regulatory effects of elevated serum Pi on PC tumorigenesis in vivo will provide crucial insights into the pathophysiological role of elevated Pi. Moreover, extending the exposure duration of elevated serum Pi in vivo could offer a more accurate understanding of how chronic Pi exposure impacts PC cell behavior.

## 4. Materials and Methods

### 4.1. Cell Line

The PLum-AD cell line, generated from murine orthotopic adenocarcinoma tumors, representing early-stage PC, was used in this study. PLum-AD cells exhibit an epithelial morphology with high CK8 and low Vim [56]. Harboring *Pten* and *TP53* gene deletions, this cell line displayed the potential for self-renewal, growth into CRPC cells, and metastasis through the EMT, as such being a cellular model of PC progression [55].

### 4.2. Cell Culture

PLum-AD PC cells were cultured in Dulbecco’s modified Eagle’s medium/Ham’s F-12 (Advanced DMEM/F-12) medium containing 1 mM Pi (Gibco), supplemented with 5 μg/mL Plasmocin^®^ Prophylactic (InvivoGen, San Diego, CA, USA), 1% penicillin–streptomycin (Sigma-Aldrich, St. Louis, MO, USA), 1% GlutaMAX (Gibco), 1% HEPES (Gibco), and 5 ng/mL of EGF (R&D Systems, Minneapolis, MN, USA). Cells were maintained in a humidified atmosphere with 5% CO_2_ at 37 °C and kept mycoplasma-free.

### 4.3. Treatment Protocols

NaPi and KPi were used as separate sources of Pi, with NaCl and KCl serving as controls, respectively. To prepare the NaPi stock solution, sodium phosphate dibasic (HNa_2_O_4_P; Sigma-Aldrich) and sodium phosphate monobasic dihydrate (H_2_NaO_4_P·2H_2_O; Sigma-Aldrich) were dissolved in double-distilled water (ddH_2_O) at a 4:1 ratio. Similarly, the KPi solution was prepared by dissolving potassium phosphate dibasic (HK_2_O_4_P; Sigma-Aldrich) and potassium phosphate monobasic (H_2_KO_4_P; Sigma-Aldrich) in ddH_2_O at a 4:1 ratio [57]. The pH of each stock solution was measured and, if necessary, adjusted to 7.4 ± 0.2 and stored at 4 °C for up to 3 weeks.

In all experiments, untreated cells cultured in a complete medium containing 1 mM Pi were used as controls reflecting normal serum levels while those cultured in 2% ddH_2_O were used as vehicle controls. Cells were treated with 1 mM NaPi or KPi to model physiologically relevant hyperphosphatemia. Severe hyperphosphatemia was modeled using 3 mM NaPi or KPi, while supraphysiological hyperphosphatemia was modeled with concentrations of 6 mM or higher.

### 4.4. Thiazolyl Blue Tetrazolium Bromide Cell Proliferation Assay

The in vitro proliferative effects of NaPi and KPi were assessed using the MTT (Sigma-Aldrich) colorimetric assay [58]. PLum-AD cells (2.5 × 10^3^ cells/well) were seeded in triplicate in 96-well plates (Corning, Corning, NY, USA) and cultured overnight. The cells were then treated with 1, 3, 6, 9, and 20 mM of NaPi and KPi and 1, 6, and 20 mM of NaCl and KCl for 24, 48, and 72 h. At each time point, 5 mg/mL of MTT reagent (Sigma-Aldrich) was added to each well and incubated at 37 °C with 5% CO_2_ for 3 h, followed by the addition of the solubilizing agent 2-propanol (Sigma-Aldrich). Optical density (OD) was measured at 595 nM using the Tristar Multimode Reader (BERTHOLD, Bad Wildbad, Germany). The proliferation percentage relative to the control was then calculated and represented.

### 4.5. Trypan Blue Exclusion Cell Viability Assay

The trypan blue exclusion assay was used to assess the effects of NaPi and KPi on cell viability [59]. PLum-AD cells (15 × 10^3^ cells/well) were seeded in duplicate in 24-well plates and cultured overnight. Cells were then treated with 1, 3, 6, and 9 mM of NaPi, KPi, NaCl, and KCl for 24, 48, and 72 h. At each time point, attached cells were harvested and stained with trypan blue solution (Sigma-Aldrich). Viable cells were counted using a hemocytometer under an Axiovert inverted light microscope from Zeiss (San Diego, CA, USA). Viability was represented as a percentage relative to the control.

### 4.6. Wound Healing Cell Migration Assay

The migratory capacity of PLum-AD cells upon treatment with NaPi and KPi was evaluated using the wound healing/scratch assay [60]. PLum-AD cells (9 × 10^4^ cells/well) were seeded in duplicate in 24-well plates and cultured overnight. To inhibit cell proliferation, cells were treated with 10 μg/mL of Mytomycin C (Sigma-Aldrich) for 30 min at 37 °C in a 5% CO_2_ humidified incubator. Then, a uniform scratch was etched in each well using a sterile 200 μL micropipette tip, after which the cells were gently washed with 1× phosphate-buffered saline (PBS) solution to remove any detached cell debris. The cells were then treated with 1 and 3 mM of NaPi and KPi. Images of the wound area in each well were captured at 0, 3, 6, 9, 12, and 14 h post-treatment using Axiovert bright-field microscopy from Zeiss at a 5× magnification. Wound area was analyzed using the ImageJ software version 1.53, and the percentages of wound closure for the different time points (t) were calculated for each condition.

### 4.7. Immunofluorescence Staining and Analysis

PLum-AD cells were seeded onto 100 × 20 mm Petri dishes and cultured overnight, followed by treatment with 1 and 3 mM of NaPi and KPi for 24 and 48 h. At each time-point, 100 μL of the cell suspension from each condition was deposited onto Starfrost slides (Knittel Glass) using a Cytospin^TM^ 4 Centrifuge (Thermo Scientific, Waltham, MA, USA). Cells were fixed with 4% paraformaldehyde (PFA) (Sigma-Aldrich), permeabilized with 0.5% Triton X-100 in PBS, and blocked with a buffer solution [0.1 % Bovine Serum Albumin (BSA), 0.2% Triton X-100, 0.05% Tween-20, and 10% Normal Goat Serum (NGS) in 1X PBS] at room temperature. The cells were stained and incubated overnight at 4 °C with the primary antibodies anti-CK8 (1/200 dilution; Santa Cruz Biotechnology, Dallas, TX, USA) and anti-Vim (1/200 dilution; Santa Cruz Biotechnology). On the second day, the cells were incubated for 1 h with the secondary antibodies Alexa 488 and 568 conjugated IgG (1/200 dilution; Thermo Fisher Scientific). Slides were then mounted with 4′,6-diamidino-2-phenylindole (DAPI) for nuclear staining, and fluorescent signals were captured from four random representative fields using a Leica DM6 B Fully Automated Upright Microscope with a 20× objective. Quantification of CK8 and Vim expression was performed manually as a percentage of DAPI-stained cells.

### 4.8. Statistical Analysis

Statistical analysis was performed and data graph plots were compiled using the GraphPad Prism 9 software version 9.5.1 (GraphPad Software, Palo Alto, CA, USA). Data are expressed as the mean ± standard error of the mean (SEM) of at least 3 independent experiments. Differences between individual groups were analyzed using two-way ANOVA followed by Bonferroni post hoc tests for multiple comparisons. Statistical significance was reported at *p*-values < 0.05 (* *p* < 0.05, ** *p* < 0.01, and *** *p* < 0.001).

## 5. Conclusions

While epidemiological studies established an association between higher phosphorus intakes and the risk of lethal and high-grade PC, our in vitro study results suggest that elevated Pi levels *per se*, in the hyperphosphatemia range, can directly promote EMT in PC and thus may play a role in tumor progression. This finding highlights the potential role of targeting Pi in the TME of PC cells to manage advanced stages of the disease.

## Figures and Tables

**Figure 1 ijms-26-04762-f001:**
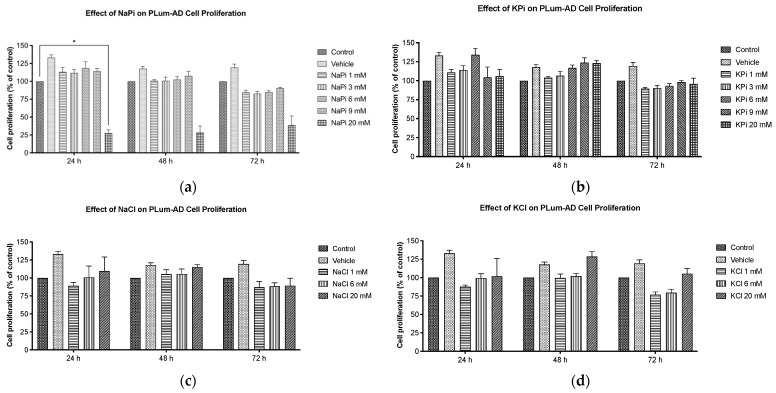
Supraphysiological elevated concentrations of NaPi inhibit cell growth of murine prostate cancer cells. The effect upon treatment with varying concentrations of NaPi (**a**), KPi (**b**), NaCl (**c**), and KCl (**d**) on PLum-AD cell proliferation was assessed using the MTT assay. PLum-AD cells were seeded in triplicate in 96-well plates and incubated for 24, 48, and 72 h with or without the indicated concentrations of the treatment. Results are expressed as the percentage of cellular proliferation of the treated group compared to the control at each time point. Data represent an average of three independent experiments, reported as mean ± SEM and analyzed using two-way ANOVA. Statistical significance reported at *p*-values < 0.05 (* *p* < 0.05). MTT, 3-(4, 5-dimethylthiazol-2-yl)-2, 5-diphenyltetrazolium bromid; NaCl, sodium chloride; NaPi, sodium phosphate; KPi, potassium phosphate; KCl, potassium chloride.

**Figure 2 ijms-26-04762-f002:**
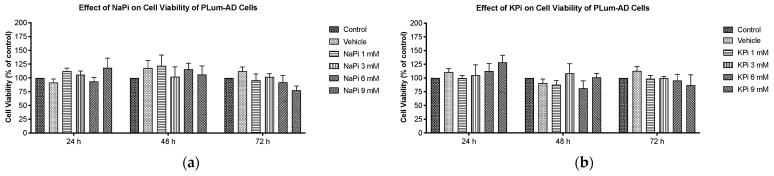
Increased concentrations of NaPi and KPi exert no effect on the cell viability of murine prostate cancer cells. PLum-AD cells were seeded in duplicate in 24-well plates. After incubation for 24, 48, and 72 h with or without the indicated treatment concentrations, cell viability was assessed using the trypan blue exclusion assay. Results are expressed as a percentage of viability of the NaPi (**a**), KPi (**b**), NaCl (**c**), and KCl (**d**) groups compared to controls at 24, 48, and 72 h time points. Data represent an average of four independent experiments, reported as mean ± SEM and analyzed using two-way ANOVA. Statistical significance reported at *p*-values < 0.05 (* *p* < 0.05 and ** *p* < 0.01). NaCl, sodium chloride; NaPi, sodium phosphate; KPi, potassium phosphate; KCl, potassium chloride.

**Figure 3 ijms-26-04762-f003:**
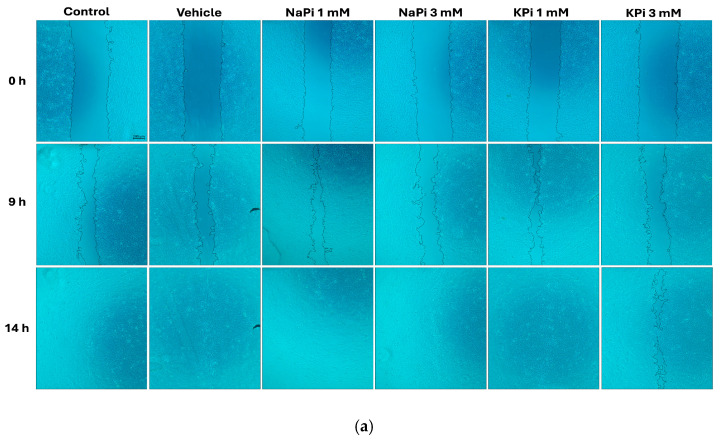
Elevated concentrations of NaPi and KPi exert no effect on the migratory ability of murine prostate cancer cells. PLum-AD cells were seeded in duplicate in 24-well plates. A uniform scratch was etched on confluent cells using a 200 μL tip and images were then taken at 0, 3, 6, 9, 12, and 14 h with or without the indicated treatment concentrations. Representative images at 5× (scale bar = 200 μm) at 0, 9, and 14 h, analyzed using ImageJ software version 1.53, are shown (**a**). Quantification of the open wound area was performed over time using the ImageJ software. Results are expressed as a percentage of open wound area of the NaPi-treated groups (**b**) and KPi-treated groups (**c**) compared to the control at each time point. Data represent an average of three independent experiments, reported as mean ± SEM and analyzed using two-way ANOVA. NaPi, sodium phosphate; KPi, potassium phosphate.

**Figure 4 ijms-26-04762-f004:**
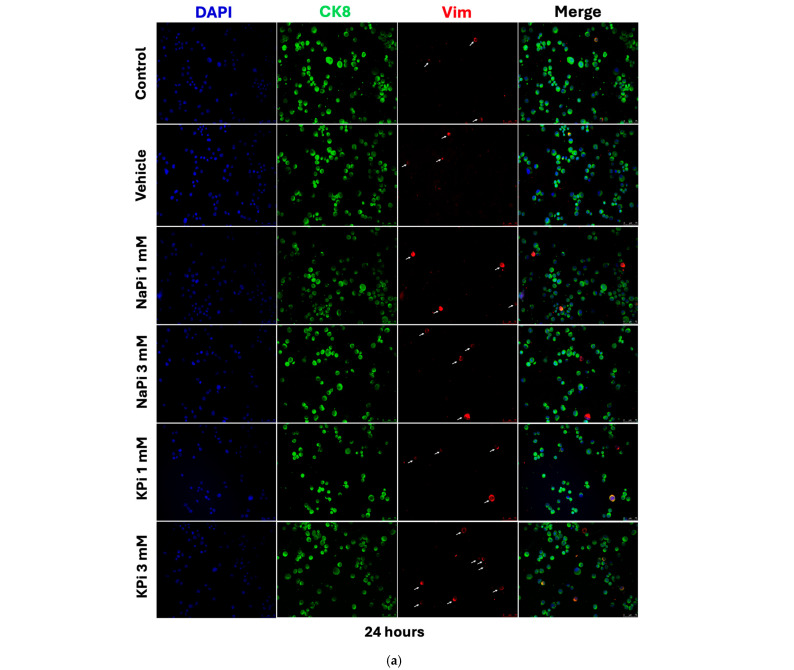
Elevated concentrations of NaPi and KPi increased the relative expression of vimentin in murine prostate cancer cells. Representative immunofluorescent images of PLum-AD cells stained for EMT markers (epithelial CK8 and mesenchymal Vim) and the nuclear counterstain DAPI are shown at 20× (scale bar = 75 μm) for time points 24 (**a**) and 48 h (**b**) with or without treatment. White arrows denote the location of Vim positive cells (**a**,**b**). Cells positively stained for Vim after treatment with 1 and 3 mM NaPi (**c**) and KPi (**d**) for 24 and 48 h were plotted as percentages of the total number of positively stained cells for DAPI per experiment. Data represent an average of three independent experiments, reported as mean ± SEM and analyzed using two-way ANOVA. Statistical significance reported at *p*-values < 0.05 (* *p* < 0.05, *** *p* < 0.001). DAPI, 4′,6-diamidino-2-phenylindole; CK8, cytokeratin 8; NaPi, sodium phosphate; KPi, potassium phosphate; Vim, vimentin.

## Data Availability

The data supporting the findings of this study are available from the corresponding authors upon request.

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
