# Peer review of "Effect of Inorganic Phosphorus Manipulation on the Growth and Progression of Prostate Cancer Cells In Vitro"

_ijms, 2025, doi:10.3390/ijms26104762_

Round 1

Reviewer 1 Report

Comments and Suggestions for Authors

Thank you for the opportunity to review this manuscript, which investigates the effects of inorganic phosphorus (Pi) on the growth and progression of prostate cancer (PC) cells in vitro. While the study addresses an important and clinically relevant topic, several areas require clarification. Below, I provide a detailed critique of the manuscript:

Major Comments:

  1. The study demonstrates that elevated Pi levels increase vimentin expression, suggesting a role in promoting EMT. However, the underlying molecular mechanisms driving these effects are not explored. For instance, the involvement of key signaling pathways such as AKT, MAPK, and ERK1/2, which are known to regulate EMT and cancer progression, is not investigated. This is a limitation of the study that must be mentioned in the manuscript.

  1. While the in vitro model provides valuable insights, the findings need to be validated in an in vivo setting to confirm the relevance of Pi in tumor progression and metastasis. Make a topic in your work about future expectations that should be taken after the results of your research and explore the need for developing in vivo studies.
  2. Although the study includes controls with NaCl and KCl, it is not entirely clear whether the observed effects are due to Pi specifically or changes in osmolarity.

  1. The study focuses on vimentin as a marker of EMT but does not assess other key markers, such as E-cadherin, N-cadherin, or Snail.

Minor Comments:

  1. Introduction:
  • The introduction could benefit from a more detailed discussion of the potential role of Pi in cancer progression, including references to previous studies on Pi transporters (e.g., NaPi2b) and their involvement in tumorigenesis.

  1. Results:
  • The quality of figure 3 (cell migration assay) needs to be improved.

  1. Discussion:
  • The discussion could be expanded to include a more detailed comparison with previous studies on Pi and cancer progression.
  • The authors should also discuss the limitations of their study.

  1. Materials and Methods:
  • The concentration of Mytomycin C used in the wound healing assay should be specified.
  • The statistical analysis section should be expanded to include details on the number of replicates.

Author Response

Response to reviewer 1 is uploaded

Reviewer 2 Report

Comments and Suggestions for Authors

The manuscript “Effect of Inorganic Phosphorus Manipulation on the Growth and Progression of Prostate Cancer Cells In Vitro” provides a detailed investigation into how elevated extracellular phosphate levels directly influence cellular behavior in prostate cancer. The authors employ a robust experimental design using the PLum-AD cell line to assess proliferation, viability, migration, and epithelial‐mesenchymal transition (EMT) marker expression in response to both sodium phosphate (NaPi) and potassium phosphate (KPi). The utilization of standard techniques such as the MTT assay, trypan blue exclusion, wound healing assays, and immunofluorescence staining enhances the reproducibility and clarity of the findings.

The study’s data clearly demonstrate that while physiologically relevant concentrations of inorganic phosphate do not significantly alter cell proliferation or migration, they are capable of inducing an upregulation of vimentin—a hallmark of EMT. This observation supports the epidemiological evidence linking high dietary phosphorus intake to an increased risk of prostate cancer. Moreover, the discussion effectively addresses potential mechanisms, including the involvement of Na‐dependent phosphate transporters and signaling pathways such as AKT, MAPK, and ERK1/2. However, the current discussion could be further enriched by broadening the contextual framework regarding the role of various dietary elements and trace minerals in modulating prostate cancer risk.

In this regard, it is recommended that the authors consider incorporating additional literature to deepen the background discussion. For instance, the study with DOI 10.3390/nu16040527 in Nutrients could provide valuable insights into how nutritional factors—including other minerals and micronutrients—interact with phosphorus metabolism and influence prostate cancer risk. Similarly, the research reported in Cancers (DOI 10.3390/cancers16152618) offers complementary perspectives on the molecular mechanisms by which dietary components may impact tumor biology. Including these references would not only broaden the discussion but also help to elucidate potential synergistic effects between phosphorus and other dietary elements, such as calcium and vitamin D, which are known to influence prostate cancer progression.

Furthermore, while the in vitro model used in the current study is appropriate for mechanistic exploration, a discussion acknowledging its limitations in recapitulating the complexity of the in vivo tumor microenvironment would be beneficial. Future investigations employing in vivo models and extended treatment durations could validate the in vitro observations and provide further insights into the long-term effects of elevated inorganic phosphate on tumor progression. Additionally, a more detailed exploration of the downstream signaling pathways could strengthen the mechanistic link between phosphate-induced EMT and prostate cancer metastasis.

In summary, the manuscript makes a significant contribution to our understanding of how inorganic phosphate can modulate prostate cancer cell behavior. By integrating the findings of studies such as those with DOIs 10.3390/nu16040527 and 10.3390/cancers16152618, the authors could offer a more comprehensive view of the nutritional and molecular factors that influence prostate cancer risk, ultimately enhancing the overall impact and translational relevance of the work.

Author Response

Response is uploaded

Round 2

Reviewer 1 Report

Comments and Suggestions for Authors

The authors responded appropriately to the corrections requested by the reviewer. I recommend acceptance of the manuscript.